# IMPROVING COMPOSITIONAL TEXT-TO-IMAGE GENERATION WITH LARGE VISION-LANGUAGE MODELS

## ABSTRACT

Recent advancements in text-to-image models, particularly diffusion models, have shown significant promise. However, compositional text-to-image models frequently encounter difficulties in generating high-quality images that accurately align with input texts describing multiple objects, variable attributes, and intricate spatial relationships. To address this limitation, we employ large vision-language models (LVLMs) for multi-dimensional assessment of the alignment between generated images and their corresponding input texts. Utilizing this assessment, we fine-tune the diffusion model to enhance its alignment capabilities. During the inference phase, an initial image is produced using the fine-tuned diffusion model. The LVLM is then employed to pinpoint areas of misalignment in the initial image, which are subsequently corrected using the image editing algorithm until no further misalignments are detected by the LVLM. The resultant image is consequently more closely aligned with the input text. Our experimental results validate that the proposed methodology significantly improves text-image alignment in compositional image generation, particularly with respect to object number, attribute binding, spatial relationships, and aesthetic quality.

## 1 INTRODUCTION

Recently, text-to-image models have made great progress, of which diffusion models are remarkable. Compositional text-to-image generation is a more advanced task, which understands and generates images with multiple objects which have variable attributes and complex spatial relationships. Although diffusion models can generate high-quality images, compositional text-to-image generation still struggles to generate images aligned with input texts. These limitations manifest inaccuracies in object number, attribute binding, spatial relationships between objects and aesthetic quality, as shown in Figure 1. These inaccuracies are caused by the compositional complexity of the input text and the cross-attention mechanism in the diffusion model.

Several studies (Agarwal et al., 2023; Chefer et al., 2023) have attempted to resolve issues related to attribute binding by manipulating the attention mechanism within diffusion models. However, these approaches often fall short of comprehensively addressing jointly the interrelated challenges of object number, attribute binding, spatial relationships between objects, and aesthetic quality. Besides, the evaluation of the alignment of the generated image and input text is not fully explored. Several techniques, such as CLIPScore (Hessel et al., 2021) or BLIP (Li et al., 2022), fall short in capturing compositional alignment accurately. While there are methodologies like T2I-CompBench (Huang et al., 2023) that introduce compositional evaluation methods, these are primarily applied to fine-tune diffusion models during the training phase. This approach fails to exploit the full potential of compositional evaluation methods, leaving room for enhancing the overall alignment between generated images and input texts in the inference period.

Parallel to these developments, Large Language Models (LLMs), such as GPT-4, LLama, and Vicuna, have emerged as influential tools in both academia and industry. Building upon LLMs, Large Vision-Language Models (LVLMs) have been developed to integrate visual features with language representations, endowing LLMs with multimodal capabilities. Notable models such as MiniGPT-4 (Zhu et al., 2023), LLama-Adapter (Zhang et al., 2023a), and Bard (Google, 2023) have demonstrated remarkable zero-shot learning, visual perception and reasoning abilities.

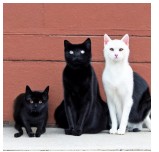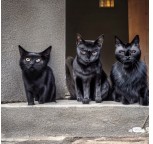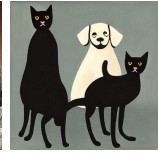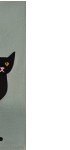 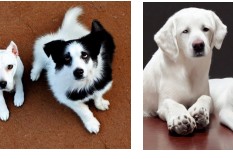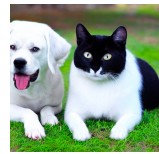

Input text: three black cats and a white dog
a. Object number

Input text: a white dog and a black cat
b. Attribute binding

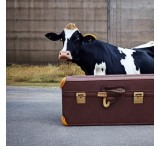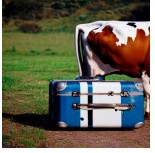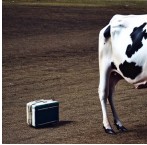 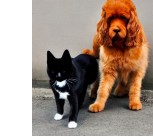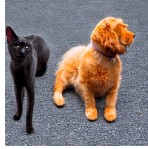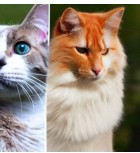

Input text: a suitcase on the right of a cow
c. Spatial relationship

Input text: a cat and a dog
d. Aesthetic quality

Figure 1: **Illustrating Limitations in Compositional Text-to-Image Generation.** (a) Object Number: The discrepancy between the quantity of objects in the image (e.g., cat and dog) and the input text is evident. (b) Attribute Binding: The attributes of objects depicted do not correspond with the input text; for instance, the cat's color is black and white, contrasting with the specified black. (c) Spatial Relationship: The arrangement of objects does not conform to the input text, with the suitcase not situated to the right of the cow as described. (d) Aesthetic Quality: The representation of the object is distorted, deviating from conventional aesthetic standards.

Leveraging these advancements, our work incorporates the LVLM to augment the capabilities of compositional text-to-image generation. Specifically, we first utilize LVLMs for evaluation. To capture the compositional feature in alignment, we evaluate the alignment of generated images with input texts mainly in terms of four dimensions: object number, attribute binding, spatial relationship, and aesthetic quality. We employ the LVLM to formulate questions derived from the input text and subsequently input the generated image and questions into the LVLM. The responses obtained serve as LVLM-assessed metrics to evaluate alignment. Following this, we employ Reward Feedback Learning (ReFL) during the training phase to fine-tune diffusion models based on these metrics, thereby enhancing compositional text-to-image generation. To fully utilize the evaluation method, during the inference stage, the LVLM is re-engaged to identify and correct errors in the generated images. An iterative process governed by the LVLM uses image-editing algorithms to eliminate misalignments until the generated image is fully compliant with the input text. Our empirical results substantiate that our approach significantly amplifies the accuracy and fidelity of compositional image generation.

In conclusion, the key contributions of our study include:

- We leverage LVLMs to assess the alignment between generated images and input texts, focusing on object number, attribute binding, spatial relationships, and aesthetic quality within compositional text-to-image models.

- We enhance image-text alignment by fine-tuning the diffusion models through LVLM-based evaluations during the training period.

- We design an LVLM-guided iterative correction process to systematically rectify any misalignments in the generated images during the inference period.

Through these contributions, our work establishes a robust and plug-and-play framework for improving compositional text-to-image diffusion models.

## 2 RELATED WORK

**Text-to-Image Generation.** The aim of text-to-image generation is to produce images based on input textual descriptions. Significant advances in generative models, such as generative adversarial networks (GANs (Goodfellow et al., 2014)), auto-regressive models (Vaswani et al., 2017), and diffusion models (Ho et al., 2020), have paved the way for a plethora of works in this domain. GANs were initially employed for this purpose Reed et al. (2016), and numerous subsequent GAN-based models sought to enhance visual fidelity and caption congruence (Zhang et al., 2017; 2018; Xu et al., 2018; Li et al., 2019; Dong et al., 2017; Zhu et al., 2019; Tao et al., 2020; Ye et al., 2021; Kang et al., 2023; Sauer et al., 2023). However, GANs are not without challenges, particularly in terms of mode-collapse and training instability.

To address these issues, researchers have investigated the use of Transformer-based auto-regressive models for text-to-image generation (Ramesh et al., 2021; Ding et al., 2021; Esser et al., 2021a; Ding et al., 2022; Zhang et al., 2021; Lee et al., 2022; Chang et al., 2023), combined with a discrete VAE (Van Den Oord et al., 2017; Razavi et al., 2019; Esser et al., 2021b) for image tokenization and Transformers (Vaswani et al., 2017) for modeling the joint distribution of textual and image tokens. This often follows a two-stage methodology: initially, a discrete VAE tokenizes the input image, and subsequently, a multi-layer Transformer integrates text and image tokens.

Diffusion models have also been embraced for text-to-image generation (Nichol et al., 2021; Ho et al., 2022; Ramesh et al., 2022; Saharia et al., 2022; Rombach et al., 2022; Xu et al., 2022; Zhang et al., 2023b). For instance, GLIDE (Nichol et al., 2021) innovatively conditions the diffusion model on an input caption, building upon earlier works in the diffusion model sphere (Dhariwal & Nichol, 2021; Ho & Salimans, 2022). Moreover, DALL-E 2 (Ramesh et al., 2022) enhances the GLIDE model by conditioning on a supplemental CLIP image embedding for heightened diversity. Some endeavors, like Stable Diffusion (Rombach et al., 2022), emphasize computational efficiency by first representing input images as low-dimension latent codes. Nevertheless, challenges such as alignment with human preferences and textual input continue to persist.

**Alignment of Text-to-Image Generation Models.** Efforts have been made to align text-to-image generation models with human preferences and aesthetic standards (Hao et al., 2022; Lee et al., 2023; Wu et al., 2023; Xu et al., 2023; Dong et al., 2023a; Fang et al., 2023). For instance, Lee et al. (2023) concentrates on text alignment, using a reward model trained on human-annotated datasets to refine the text-to-image model. Similarly, ImageReward (Xu et al., 2023) offers a universal human preference reward model that encompasses text-image alignment, body problems, aesthetics, toxicity, and biases.

Promptist Hao et al. (2022) introduces prompt adaptation by training a language model to enhance the original prompt. This method leverages both the CLIP model and an aesthetic predictor as reward models, and fine-tunes in a supervised manner within a reinforcement learning framework. Alternatively, given the inefficiencies and instabilities associated with Reinforcement Learning from Human Feedback (RLHF (Ouyang et al., 2022)), Dong et al. (2023a) proposes reward ranked finetuning to better align generative models.

Despite the strides made in aligning text-to-image generation models with human preferences and aesthetic standards, complexities still arise when dealing with intricate prompts delineating multiple objects, diverse attributes, and elaborate spatial relations. Addressing these challenges, our work innovatively integrates large vision-language models (LVLMs) with diffusion models, presenting a refined approach for enhancing text-image alignment, especially in the realm of compositional image generation.

## 3 METHOD

### 3.1 PRELIMINARY

**Latent Diffusion Models.** Recently, diffusion models, exemplified by DALL-E and Midjourney, have gained widespread adoption in the field of text-to-image generation. These generative models aim to create desired data by denoising from a Gaussian distribution $x_T \sim N(0,1)$. Initially, the diffusion models define a forward process by constructing a Markov chain of variables $x_1, x_2, ..., x_T$

from the target distribution $x_0 \sim q(x_0)$ by iteratively adding Gaussian noise based on a predefined schedule $\beta_t$:

$$q(x_t|x_{t-1}) = N(x_t; \sqrt{1 - \beta_t}x_{t-1}, \beta_t I). \tag{1}$$

Subsequently, the target distribution is transformed to a Gaussian distribution at step $T$. The diffusion models are tasked with learning the reverse process to approximate the true posterior $q(x_{t-1}|x_t)$ by denoising from Gaussian distribution $x_T \sim \mathcal{N}(0, 1)$:

$$p_\theta(x_{t-1}|x_t) = \mathcal{N}(x_{t-1}; \mu_\theta(x_t, t), \sigma_t), \tag{2}$$

where $\mu_\theta$ represents the mean, computed using neural networks. This reverse process generates the desired sample $x_0$ at last. In contrast, latent diffusion models execute the aforementioned forward and reverse processes in the latent space rather than the pixel space. This adaptation aims to mitigate the computational cost and enhance semantic generation. These models employ the Variational Autoencoder (VAE) to encode to the latent space $z_0 = f(x_0)$. They can be used in text-to-image generation by inputting a prompt $T$ to generate the corresponding image:

$$p_\theta(z_{t-1}|z_t) = \mathcal{N}(z_{t-1}; \mu_\theta(z_t, t, T), \sigma_t). \tag{3}$$

The training loss is derived from the variational lower bound (VLB) loss, which can be simplified as

$$\mathcal{L}(\theta) = E_{t\sim[1,T],z_0,\epsilon_t}[||\epsilon_t - \epsilon(z_t, t, T)||_2^2]. \tag{4}$$

Stable Diffusion is based on latent diffusion models, which allows latent diffusion models to be versatile and efficient in generating high-quality, semantically coherent images from textual prompts. This method serves as the baseline for our study.

**ImageReward.** Latent Diffusion Models (LDMs) have shown significant potential as generative models. One of the highlighted challenges in the realm of LDMs is their direct optimization. The ReFL (Xu et al., 2023) methodology, as discussed in prior works, offers a potential solution to this optimization challenge.

LDMs follow a sequential denoising process, which in experiments, expands up to 40 steps. A significant observation from the ReFL approach highlights the behavior of model scores during these denoising steps:

- **Early Stages (Steps 1 to 15):** In this phase, the model scores remain uniformly low for all generated outputs.
- **Intermediate Stages (Steps 15 to 30):** Here, while high-quality generations become evident, it's still early to conclusively gauge the final quality of all generations based on the present model scores.
- **Late Stages (Steps 30 onwards):** At this juncture, there's a discernible distinction in generations based on their respective model scores.

These observations suggest that model scores after the 30th denoising step could be potentially reliable indicators for enhancing LDMs, even if they aren't derived from the final step.

The ReFL algorithm, as elucidated in the referenced literature, aims to harness these model scores as feedback to back-propagate and refine the LDMs. This stands in contrast to traditional methodologies where only the gradient from the final denoising step is retained – an approach found to yield instability.

For effective fine-tuning, there is a balance established between the ReFL loss and the pre-training loss, a strategy to counter rapid overfitting and ensure a more stable fine-tuning process. The corresponding loss functions are illustrated as:

$$\mathcal{L}_{reward} = \lambda \mathbb{E}_{y_i \sim \mathcal{Y}}(\phi(r(y_i, g_\theta(y_i)))) \tag{5}$$

$$\mathcal{L}_{pre} = \mathbb{E}_{(y_i, x_i)\sim\mathcal{D}}(\mathbb{E}_{\mathcal{E}(x_i), y_i, \epsilon\sim\mathcal{N}(0,1),t}[||\epsilon - \epsilon_\theta(z_t, t, \tau_\theta(y_i))||_2^2]) \tag{6}$$

Here, $\theta$ denotes the parameters of the LDM, and $g_\theta(y_i)$ represents the generated image of the LDM using parameters $\theta$ corresponding to prompt $y_i$. Our approach synergistically combines large vision-language models with diffusion models, wherein the LVLM evaluates the generation outcomes, subsequently guiding the ReFL training of the diffusion model. This integration of LVLMs and diffusion models via ReFL underpins a pioneering methodology for enhancing text-image coherence, especially for intricate prompts. This strategy is pivotal to our research and establishes the advanced framework upon which our study is built.

## 3.2 OVERVIEW

In this study, we introduce a comprehensive framework that leverages LVLMs to enhance compositional text-to-image generation. The Figure 2 provided illustrates a schematic representation of our proposed framework, which integrates three core components: LVLM-based Evaluation, Model Fine-tuning, and LVLM-guided Image Editing. Each component strategically utilizes the capabilities of LVLMs to optimize the generation process.

Initially, we deploy the LVLM to assess the alignment between generated images and input texts. The LLMs analyze the input texts and formulate questions aimed at capturing compositional features inherent in the texts. The generated images, along with these questions, are then fed into the LVLM, which produces answers serving as evaluative metrics for alignment assessment. Subsequent to the LVLM-based evaluation, we employ Reward Feedback Learning (ReFL) to fine-tune the diffusion models. This process aims to optimize the models based on the evaluative metrics derived from LVLM evaluation during the training period, thereby enhancing alignment between the images generated and the input texts. To fully harness the potential of the LVLM, we incorporate it during the inference stage as well. Specifically, the LVLM is used to identify any misalignment between the generated images and the input texts. Upon detection of misalignments, the LVLM is used to guide the correction process with image-editing algorithms until the generated images are fully aligned with the input texts, ensuring no misalignment remains.

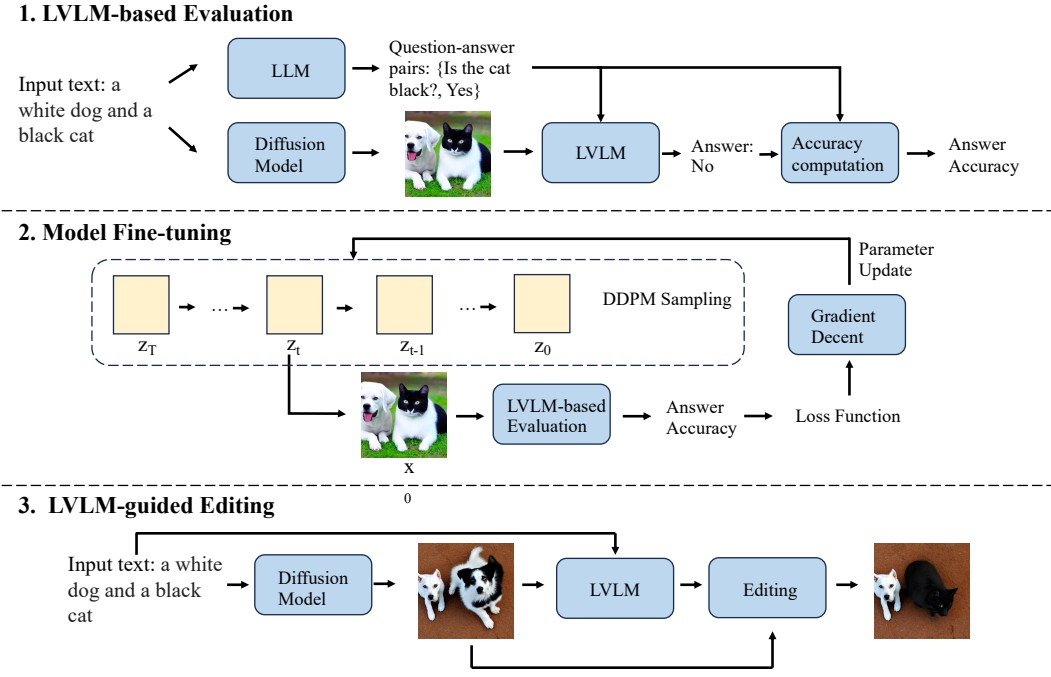

Figure 2: **Overview of the Proposed Methodology.** Our methodology is structured around three core components: (1) LVLM-based Evaluation: Drawing inspiration from TIFA, we initially employ LLM to formulate question-answer pairs grounded in the input text. Subsequently, the LVLM is utilized to procure answers by processing the formulated questions alongside the image. A comparative analysis of answers derived from both image and text is then undertaken to calculate the answer accuracy, serving as our evaluative metric. (2) Model Fine-tuning: The LVLM-based evaluation metric is incorporated as a weight within the diffusion loss function, facilitating the fine-tuning of the diffusion model. The objective is to guide the diffusion model's focus towards enhancing answer accuracy. (3) LVLM-guided Editing: In the inference phase, the LVLM is deployed to identify misalignments between image and text. Subsequent to this identification, image-editing algorithms are applied iteratively to rectify the image until no alignment is detected.

### 3.3 LVLM-BASED EVALUATION

To evaluate the alignment of generated images in compositional text-to-image synthesis, we integrate Large Vision-Language Models (LVLMs) into the Text-to-Image Faithfulness evaluation with Question Answering (TIFA (Hu et al., 2023)) framework. This framework assesses various elements such as objects, shapes, materials, attributes, and spatial relationships in the images by formulating specific prompts. The evaluation process unfolds as follows:

**Text Analysis.** The initial step involves analyzing the input text $T$. In adherence to TIFA guidelines, we employ LLMs to generate question-answer pairs $\{Q_i, A_i\}_{i=1}^N$ derived from the input text $T$, thereby utilizing the zero-shot, in-context, and reasoning capabilities of LLMs. These generated pairs encapsulate the compositional information present in the text, encompassing aspects like object number, attribute binding and spatial relationship.

**LVLM-based Question Answering:** Based on the TIFA framework, the LVLM is used to answer the formulated questions. Upon generating question-answer pairs $Q_i, A_{i_{i=1}}^N$ from the given text $T$, both the questions $Q_i$ and the generated image $I$ are input into the LVLM. The LVLM, through reasoning on the generated image $I$, produces the answers $\tilde{A}_i$.

**Accuracy Computation:** For each pair of text and image $T, I$, we compare the answers derived from text $Q_i$ and the image $\tilde{Q}_i$. The accuracy is computed as:

$$\text{ACC}(T, I) = \sum_{i=1}^n \mathbb{1}[Q_i = \tilde{Q}_i]. \tag{7}$$

By doing so, the LVLM facilitates a nuanced and detailed evaluation in terms of the answer accuracy of how well the generated images align with the input text, examining the fidelity of the representation across object number, attribute binding and spatial relationship.

### 3.4 MODEL FINE-TUNING

To refine the alignment between text and image within diffusion models, we adopt a strategy known as Reward Feedback Learning (ReFL (Xu et al., 2023)). ReFL is employed to fine-tune diffusion models based on the LVLM-based evaluation during the training phase. The rewards in ReFL are used to backpropagate and update the diffusion parameters after a predetermined range of steps, because the latter steps yield clearer images, conducive for use in the LVLM, thereby enhancing the stability of the training process. During our model fine-tuning, we first sample plenty of text-image pairs from the diffusion model, subsequently employing the previously mentioned answer accuracy to assess each pair. However, the LVLM-based evaluation is non-differential. Different from ReFL, the answer accuracy serves as the weight of the loss function for fine-tuning the diffusion model. Higher answer accuracy indicates improved alignment between image and text, thus the optimization process prioritizes these instances. The loss function is formulated as:

$$\mathcal{L}'(\theta) = E_{(T,I)}[\text{ACC}(T, I) \cdot ||\epsilon - \epsilon(z_t, t, T)||_2^2], \tag{8}$$

where $(T, I)$ represents a sample of text-image pairs generated from the diffusion model, and $\text{ACC}(T, I)$ is the answer accuracy derived from LVLM-based evaluation. The algorithm is depicted in Algorithm 1.

After the fine-tuning on sampled image-text pairs, the diffusion model is optimized with a focus on enhancing the answer accuracy in subsequently generated text-image pairs.

### 3.5 LVLM-GUIDED EDITING

To maximize the utility of LVLM-based evaluation, we incorporate it not only during the training period but also in the inference period. We note that despite the significant improvement in the alignment between text and image through fine-tuning, discrepancies can still arise during inference. To address this, we iteratively correct any misalignment in the initially generated image during the inference phase. For example, if the LVLM identifies a color discrepancy between an object in the generated image and the corresponding input text, an editing algorithm is activated to adjust the object's color to align with the text description. In this process, we first generate an initial

---

**Algorithm 1** Training Procedure

---

**Input:** Fine-tuning text $\{T_j\}_{j=1}^n$, text-image pairs sampled from datasets $\{\tilde{T}_j, \tilde{I}_j\}_{j=1}^n$, number of
   diffusion step $T$, step range $[t_1, t_2]$, diffusion model $D_\theta$
 1: **for** $j = 1, ..., n$ **do**
 2:    Compute $\mathcal{L}(\theta, \tilde{T}_j, \tilde{I}_j)$ based on Eq. 4
 3:    $\theta \leftarrow \theta + \alpha_1 \frac{\mathcal{L}(\theta)}{\theta}$
 4:    $t \leftarrow \text{Random}(t_1, t_2)$
 5:    $z_t \sim \mathcal{N}(0, 1)$
 6:    **for** $i = T, ..., t + 1$ **do**
 7:       $z_{i-1} \leftarrow D_\theta(z_i)$
 8:    **end for**
 9:    $z_{i-1} \leftarrow D_\theta(z_i)$
10:    $z_0 \leftarrow z_0(z_{i-1})$
11:    $x_0 \leftarrow \text{VAE Decoder}(z_0)$
12:    Compute $\text{ACC}(x_0, I_j)$ based on Eq. 7 and $\mathcal{L}'(\theta)$ based on Eq. 8
13:    $\theta \leftarrow \theta + \alpha_2 \frac{\mathcal{L}'(\theta)}{\theta}$
14: **end for**
15: **return** network parameters $\theta$

---

image from the diffusion model, which has been fine-tuned using ReFL. Subsequently, the LVLM is employed to pinpoint instances of misalignment between text and image, such as disparities in object number, attribute binding, spatial relationships and aesthetic quality. Upon identification of misalignments, the LVLM is utilized to guide the process of rectifying the discrepancies by image-editing algorithms until alignment is achieved. Throughout this process, we leverage the editing capabilities of diffusion models. Initially, we employ the Segment Anything Model (SAM) to isolate all objects and backgrounds in the initial image. Following this, a diffusion-based inpainting model is used to modify the relevant objects or the background to achieve congruence with the input text. We delineate the correction of four types of misalignment as follows:

**Object Number.** When the LVLM discerns a discrepancy in the object number in the initial image compared to the input text, it categorizes the variance as either excess or deficit. If the object count exceeds the text description, SAM identifies specific objects, and the inpainting model eliminates the surplus entities. Conversely, if there are fewer objects, SAM targets the background, and the inpainting model introduces additional objects in the background.

**Attribute Binding.** The LVLM identifies instances where an object's attributes do not align with the descriptions provided in the input text. For instance, if the input text describes "a white dog" while the image depicts a black dog, the LVLM recognizes the color inconsistency. Subsequently, the SAM and LVLM are used to generate a mask for the incorrectly colored dog, and a painting algorithm is employed to replace the black dog with a white one, thus ensuring attribute coherence.

**Spatial Relationship.** When discrepancies in the spatial relationships among multiple objects are identified, the SAM and LVLM are utilized to select the mask of the incorrectly positioned object. Subsequently, we employ the inpainting algorithm to first remove the mislocated object, followed by adding it back in a location that is in alignment with the input text.

**Aesthetic Quality.** If the LVLM identifies that an object within the image is distorted and thus falls short of human performance standards, our initial step is to employ SAM for segmenting the distorted object. Subsequently, an inpainting algorithm is utilized to substitute the distorted object with a normalized version. This approach leverages the capability of the diffusion model, which finds generating singular objects to be a more manageable task.

## 4 EXPERIMENT

In this section, we conduct experiments to validate our proposed method. Initially, we outline the implementation specifics in Section 4.1. Subsequently, we visualize the results from the LVLM-based

Evaluation and LVLM-based Editing in Section 4.2. Additionally, we also conduct comparative assessment between the baseline and the fine-tuned model in Section 4.2.

## 4.1 IMPLEMENTATION DETAIL

**LVLM-based Evaluation.** In alignment with the TIFA approach, we employ LLama2 to generate question-answer pairs. We then use the Bard, a state-of-the-art LVLM proposed by Google, to process questions alongside the generated image, thereby producing the answers.

**Model Fine-tuning.** Adhering to the ReFL methodology, we source text-image pairs from LAION-5B and extract input text from DiffusionDB for training data. The model is fine-tuned in half-precision with a learning rate set to $10^{-5}$, while maintaining a batch size of 64 for each. The sample step range $[T_1, T_2]$ is defined as [1, 10]. A key distinction in our approach is that we do not directly optimize the answer accuracy, as it is non-differential. Instead, we focus on optimizing a weighted diffusion loss function, where the answer accuracy serves as the weight.

**LVLM-based Editing.** Bard is once again employed as the guiding LVLM for the image editing process. To rectify any misalignment identified by the Bard, we amalgamate the capabilities of SAM and the Blended Diffusion model, utilizing them as the image-editing algorithm.

## 4.2 EXPERIMENTAL RESULTS

**LVLM-based Evaluation.** We utilize Bard to generate answers by inputting both questions and the corresponding image. Illustrative examples of this process are presented in Figure 3, showcasing instances where images are generated based on the input text "a black dog is standing on a beach."

**Prompt:** Answer the multiple-choice question. Question: what animal is in the picture? Choices: dog, cat, bird, fish

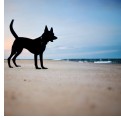

**Bard:**
The animal in the picture is a dog.
The image shows a silhouette of a dog standing on a beach. The dog has four legs, a tail, and a pointed head. This is characteristic of many dog breeds, and the overall shape of the dog in the picture is consistent with a dog.
The other choices are less likely. Cats have four legs and a tail, but they typically have a more rounded head and body. Birds have two legs and wings, and fish have fins and no legs.

**Prompt:** Answer the multiple-choice question. Question: what animal is in the picture? Choices: dog, cat, bird, fish

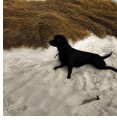

**Bard:**
Answer: dog
The image shows a black dog lying on a sandy beach. The dog has a short coat, a long tail, and floppy ears. Its head is resting on its paws, and it looks like it is enjoying the sun.
I am confident that the animal in the picture is a dog because I have seen many pictures of dogs before, and this dog has all of the physical characteristics of a dog. Additionally, the dog is lying in a position that is typical of dogs, and it looks relaxed and happy.

**Prompt:** Answer the multiple-choice question. Question: what color is the dog? Choices: black, white, red, blue

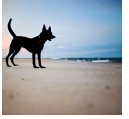

**Bard:**
The dog is black.

**Prompt:** Answer the multiple-choice question. Question: what color is the dog? Choices: black, white, red, blue

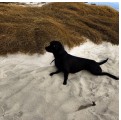

**Bard:**
The dog is black.

Figure 3: The answers produced by Bard. The images are generated with the text "a black dog is standing on a beach".

**Model Fine-tuning.** A comparative analysis is conducted between the results derived from Stable Diffusion and those from the fine-tuned model on the dataset from T2ICompBench. The visual representation of these results can be observed in Figure 4. The CLIPScore attributed to the fine-tuned model is 0.3032, which is larger than the 0.3010 score associated with Stable Diffusion.

**LVLM-based Editing.** Employing Bard, we identify misalignments within the generated images and subsequently utilize SAM and Blended Diffusion to rectify these discrepancies. Illustrative examples of this process are presented in Figure 5.

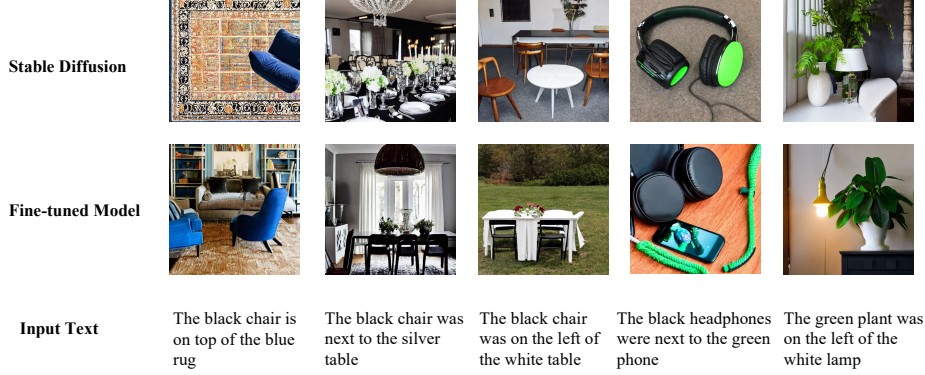

| Stable Diffusion | Fine-tuned Model | Input Text |

Figure 4: The images generated by Stable Diffusion and the fine-tuned model.

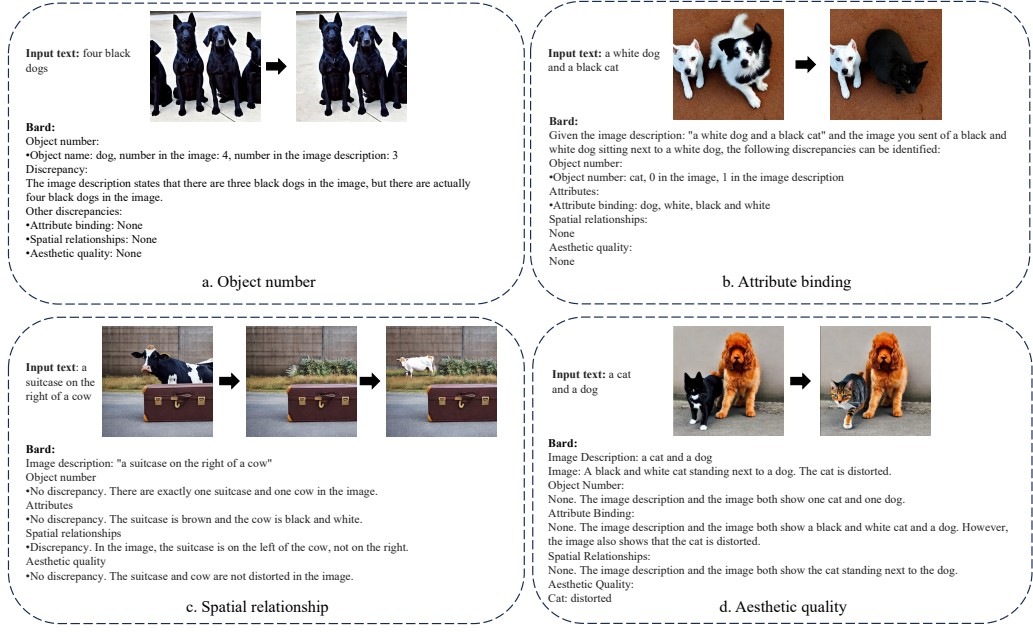

Figure 5: Visualization of LVLM-guided Editing.

## 5   CONCLUSION

**Limitations.** The efficacy of our method is constrained by the performance of the LVLMs. The current version of Bard is still not very accurate. Nevertheless, given the rapid advancements in the field of LVLMs, we will refine our approach in alignment with the development of more sophisticated LVLMs.

**Conclusion.** In this paper, our primary objective is to enhance the quality of composable image generation using LVLMs. Our methodology consists of three key components. Initially, we leverage LVLMs to assess the alignment between the generated image and the input text. Following this, we fine-tune the diffusion models utilizing LVLM-based evaluations. In the subsequent inference phase, we deploy LVLMs to detect any discrepancies between the text and the image, and an image-editing algorithm is engaged to amend these misalignments. Our empirical investigations substantiate the effectiveness of our approach in improving compositional image generation.

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

## A    ADDITIONAL RELATED WORK

**Large Vision-Language Models (LVLMs).**    Driven by the increasing diversity of large-scale data, developing powerful LVLMs has gained significant attention and progress in recent years. Early efforts, such as CLIP (Radford et al., 2021), ALIGN (Jia et al., 2021), and follow-up works (Li et al., 2021; Dong et al., 2023b; Zhang et al., 2022), adopt vision-language contrastive pre-training on extensive web-scale data, emerging superior generalization performance for zero-shot evaluation. With the popularity of large language models (LLMs) (OpenAI, 2023a;b), recent LVLMs tend to incorporate pre-trained LLMs with visual understanding capabilities. With advanced training strategies, BLIP series (Li et al., 2022; 2023) learn a Q-Former network to bridge between frozen image encoders and LLMs, which exhibit robust visual reasoning power. Trained by image-text interleaved data, Flamingo (Alayrac et al., 2022) obtains impressive few-shot learning capacity and enriches the display form of vision-language reasoning.

In contrast to the powerful but close-source GPT-4 (OpenAI, 2023b) and Bard (Google, 2023), a new branch of LVLMs is based on the open-source LLaMA (Touvron et al., 2023), and endows it with image understanding ability by visual instruction tuning. Therein, LLaMA-Adapter series (Zhang et al., 2023a; Gao et al., 2023; Han et al., 2023) introduce zero-initialized attention mechanisms, and conduct multi-modal parameter-efficient fine-tuning. LLaVA (Liu et al., 2023) introduces a high-quality visual instruction dataset to fully fine-tune the entire LLaMA, while MiniGPT-4 (Zhu et al., 2023) only adopts a projection layer for vision-language alignment. There are also many inspiring LVLM works for exploring different tuning strategies (Ye et al., 2023), collecting more diverse datasets (Chen et al., 2023), and incorporating multi-modality (Guo et al., 2023).

In this paper, as the first work, we leverage the robust vision-language reasoning of LVLMs to enhance compositional text-to-image generation. We select Bard developed by Google to first provide answers based on visual questioning, and then point out the misalignment between text prompts and generated images. Experiments have shown the effectiveness of our approach for unleashing the potential of LVLMs for improving compositional text-to-image generation.

## B    ADDITIONAL VISULIZATION

A more detailed comparison of the results between Stable Diffusion and the fine-tuned model is illustrated in Figure 6, where the images generated from the fine-tuned model exhibit a higher degree of alignment with the input text.

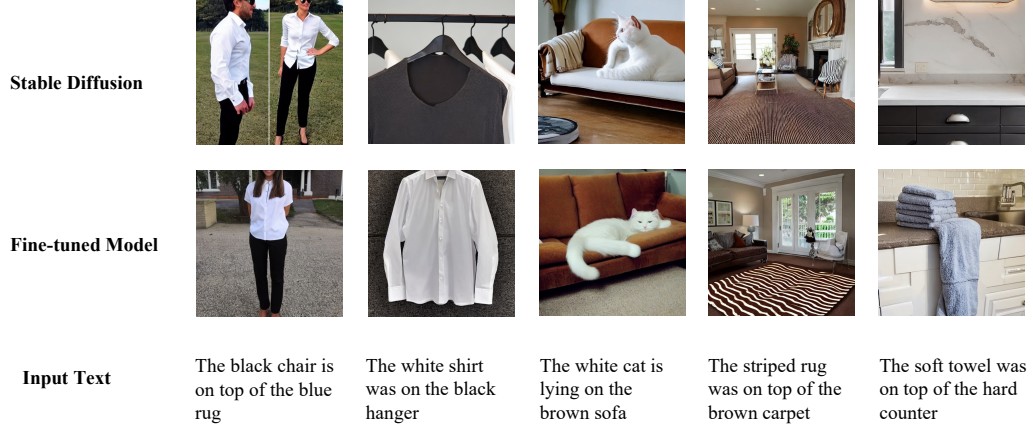

Figure 6: The images generated by Stable Diffusion and the fine-tuned model.

