# OpenReview forum: "Improving Compositional Text-to-image Generation with  Large Vision-Language Models"
_ICLR.cc/2024/Conference — Submitted to ICLR 2024_

### Official Review · Reviewer_dXyt · 2023-10-25

**Soundness:** 2 fair
**Presentation:** 3 good
**Contribution:** 2 fair
**Rating:** 5
**Confidence:** 3

**Summary:**

This paper enhances image-text alignment by fine-tuning the diffusion models through LVLM based evaluations during the training period.
and designs an LVLM-guided iterative correction process to systematically rectify any misalignments in the generated images during the inference period.

**Strengths:**

This paper establishes a robust and plug-and-play framework for improving compositional text-to-image diffusion models.

**Weaknesses:**

1. Lack of comparative experiment
2. Lack of critical ablation experiments.

**Questions:**

1、In LVLM-guided Editing,  the author introduce a image-editing algorithm which is applied iteratively to rectify the image until no alignment is detected.  Authors should provide corresponding ablation experiments without Model Fine-tuning. In other words, such method should also work directly on pre-trained Stable Diffusion, and the authors should prove it.

2、 The author introduces four limitations in  Compositional Text-to-Image Generation: (a)  Object Number (b) Attribute Binding (c) Spatial Relationship (d) Aesthetic Quality. However,  I think these limitations have been addressed to some degree in Instructpix2pix [1], Prompt-to-Prompt [2] and the authors lack relevant comparative experiments

[1] Brooks T, Holynski A, Efros A A. Instructpix2pix: Learning to follow image editing instructions[C]//Proceedings of the IEEE/CVF Conference on Computer Vision and Pattern Recognition. 2023: 18392-18402.
[2] Hertz A, Mokady R, Tenenbaum J, et al. Prompt-to-Prompt Image Editing with Cross-Attention Control[C]//The Eleventh International Conference on Learning Representations. 2022.

---

### Official Review · Reviewer_A5xo · 2023-10-25

**Soundness:** 1 poor
**Presentation:** 2 fair
**Contribution:** 2 fair
**Rating:** 3
**Confidence:** 5

**Summary:**

This paper proposes a pipeline to address the well-known problem of text-to-image generation—compositional T2I generation. This paper analyzes the object number, attribute binding, spatial relationship, and aesthetic quality problem and proposes to employ LLM to solve this problem.

**Strengths:**

This paper tries to address a very important problem—compositional text-to-image generation, which is a notorious problem in image generation area. This paper proposes a framework that employs LLM to deal with this issue. Some subjective and objective experiements show its effectiveness.

**Weaknesses:**

1. The method proposed in this paper lacks novelty. It seems like a combination of different components—a large vision-language model (e.g. llama, bard), a Reward Feedback Learning (e.g. ImageReward) and an image segmentation module (e.g. SAM). Actually, the framework is not so hard to come up with, and the overall three processes are extremely time-consuming since all the large models (language, vision, or multimodal) are huge and prohibitively expensive to implement. The contribution of this paper is not so clear.

2. Since there exist many compositional text-to-image generation algorithms based on LLM [1] or not [2, 3, 4], some of which are mentioned by you in the introduction section, no further discussion or comparisons are provided in the experimental results part. I can see only some subjective cases and a CLIPScore value, as in the experiments. You must spend more space discussing the algorithm complexity and performance comparisons with the SOTA methods.


[1] Lian, Long, et al., "LLM-grounded Diffusion: Enhancing Prompt Understanding of Text-to-Image Diffusion Models with Large Language Models." arXiv preprint arXiv:2305.13655 (2023).

[2] Chefer, Hila, et al. "Attend-and-excite: Attention-based semantic guidance for text-to-image diffusion models." ACM Transactions on Graphics (TOG) 42.4 (2023): 1-10.

[3] Feng, Weixi, et al. "Training-free structured diffusion guidance for compositional text-to-image synthesis." arXiv preprint arXiv:2212.05032 (2022).

[4]Wang, Ruichen, et al. "Compositional text-to-image synthesis with attention map control of diffusion models." arXiv preprint arXiv:2305.13921 (2023).

**Questions:**

Please refer to the weakness part

---

### Official Review · Reviewer_dnr1 · 2023-11-09

**Soundness:** 2 fair
**Presentation:** 2 fair
**Contribution:** 2 fair
**Rating:** 3
**Confidence:** 5

**Summary:**

The paper attempts to integrate various large models, including LLM (LLaMA), LVLM (Bard), a Diffusion Model (Stable Diffusion), and an Editing Model (Blended Diffusion), to establish a finetuning and inference pipeline aimed at enhancing the capability of compositional text-to-image generation. The proposed pipeline is structured in three distinct stages: LVLM-based Evaluation, Diffusion Model Finetuning, and LVLM-guided Editing. While the paper provides examples of the pipeline’s application, it relies solely on qualitative evaluation methods.

**Strengths:**

The concept of integrating several sophisticated models into a single pipeline that potentially complement each other's functionalities is commendable and could inspire more research in the field.

**Weaknesses:**

1. The premise of amalgamating various complex models is appealing, but the justification for such a pipeline, along with comprehensive evaluations of its efficacy, is lacking. The current results do not substantiate the effectiveness of the pipeline. Particularly, the necessity for both stage 2 (Diffusion Model Finetuning) and stage 3 (LVLM-guided Editing), which seemingly aim for the same goal, is not clear. There is a lack of clarity about the contributions of each stage. An in-depth ablation study would be valuable to discern the individual and collective impact of these stages.

2. The paper’s evaluation approach is insufficiently rigorous. The success cases are limited in number and are repeatedly used, without the support of robust quantitative analysis. Examples only contains simples cases, e.g., “three black dogs”, “a white cat and a black dog”. The scenarios presented could potentially be addressed by Blended Diffusion Editing alone, raising questions about the added value of the other stages.

3. The computational demands of the pipeline, particularly the LVLM-guided iterative refinement in stage 3, are not discussed. An analysis of the computational costs and the typical number of iterations required for image generation would be beneficial.

4. The quality of writing and the clarity of figures require improvement to enhance the paper's comprehensibility and professional presentation.

**Questions:**

1. There is a typographical error in Section 3.3 (LVLM-based Question Answering) involving the missing parenthesis around the $Q_i, A_i$ pairs.

2. There seems to be a mismatch between LLM (LLaMA) and LVLM (Bard) in the experiments, potentially leading to discrepancies in the answers and consequently, a lack of reliability in the accuracy measures. Would it not be more consistent to just use Bard, as it supports both textual and visual modalities?

3. Equation 7 mentions a term $\tilde{Q_i}$ which lacks a definition. Could you provide a clarification for this term?

4. In Figure 5, there is an inconsistency with the input text depicted in the first panel. The input text should be “three black dogs”?

5. References to Blended Diffusion and SAM appear to be missing when these terms are introduced in the paper.

6. The examples in Figure 4, such as the fifth case, seem to indicate that stage 2 is ineffective. This observation supports my concerns regarding the relative impact of stage 2.

Recommendation:

Given the aforementioned concerns, particularly the lack of robust evaluation and the unclear contribution of each pipeline stage, I recommend that the paper be revised before consideration for publication. The authors should address the weaknesses and questions detailed above, ensuring that the pipeline's design and efficacy are convincingly demonstrated through comprehensive experiments and quantitative assessments.

---

### Official Review · Reviewer_TYf6 · 2023-11-10

**Soundness:** 2 fair
**Presentation:** 2 fair
**Contribution:** 2 fair
**Rating:** 3
**Confidence:** 5

**Summary:**

Three main components for this paper:

Use VLM to evaluate image-text alignment by breaking down text into multiple QA

Finetune diffusion models with image-text pair that are weighted by score predicted by above methods

Test time editing (e.g., add/remove a object via inpainting) by signal from VLM

**Strengths:**

The writing is clear and the idea of the whole framework is straightforward

**Weaknesses:**

The paper does not have novelty and lacks experiment study

1, For their evaluation method, the idea of evaluation alignment by breaking down text into QA and then calling external model is first proposed by TiFA. This paper simply uses a different model (Bard) which I don't think there is anything new here.

2, for the score-weighted finetuning, it is very like: "Aligning Text-to-Image Models using Human Feedback by Kimin et al". How is it different than this paper's finetuning idea (The first part of Eq2 in Kimin's paper)?

3, I love the third component to improve alignment by editting. However, they just breifly mentioned the high-level pipeline. For exmaple, to solve incorrect attribute, they just give the high-level idea: detect the wrong object and then use inpainting method to regenerate a correct one. This is just a system, however, in a research paper, they should've include more study or in-depth analysis. For example, how accurate is the VLM model? do we need to finetune the model? if so, how to get data? Any insignful observation from prompting? etc.

4. This paper does not have a proper experiment section to compare with baselines, or ablation, and it feels like the effort was half-hearted.


I suggest the authors identify one aspect of the problem and conduct a proper and in-depth study of that issue, rather than giving a general description of a system

**Questions:**

NA

---

### Meta-Review · Area_Chair_WKRv · 2023-12-13

**Metareview:**

**Scientific Claims and Findings**:

The paper proposes a pipeline for compositional text-to-image generation, aiming to address challenges in object number, attribute binding, spatial relationship, and aesthetic quality. The pipeline integrates various large models, including a large language model (LLM), a large vision-language model (LVLM), a diffusion model (Stable Diffusion), and editing models (SAM and Blended Diffusion). The three main stages of the pipeline are LVLM-based evaluation, diffusion model fine-tuning, and LVLM-guided editing. The goal is to enhance the capability of compositional text-to-image generation through iterative refinement and alignment correction.

**Strengths of the Paper**:

- Conceptual Integration: The paper attempts to integrate various large models into a single pipeline for compositional text-to-image generation, potentially inspiring further research in the field.

- Subjective and Objective Experiments: The paper includes subjective and objective experiments demonstrating the effectiveness of the proposed framework, particularly in addressing compositional text-to-image generation challenges.

- Clear Writing and Presentation: Reviewers generally commend the clarity of writing and straightforward presentation of the pipeline's concept.

**Weaknesses of the Paper**:

- Lack of Novelty: Reviewers express concerns about the lack of novelty, with similarities to existing works and a perceived lack of clear contributions.

- Unclear Contributions of Pipeline Stages: There is a lack of clarity regarding the contributions of each stage in the proposed pipeline. Reviewers highlight the need for an in-depth ablation study to understand the individual and collective impact of these stages.

- Insufficient Experimental Study and Inadequate Evaluation Rigor: The paper is criticized for a lack of comprehensive experimental study, including limited evaluations, a dearth of comparisons with state-of-the-art methods, and the absence of critical ablation experiments. Reviewers criticize the paper for the insufficient rigor of its evaluation approach. Limited success cases and a focus on simple scenarios raise questions about the pipeline's efficacy and added value.

- Computational Demands Not Discussed and Missing Algorithm Complexity Discussion: The paper does not adequately discuss the computational demands of the proposed pipeline, especially the iterative refinement in LVLM-guided editing. Reviewers suggest the need for more discussion on algorithm complexity and performance comparisons with existing methods.

- Comparison with State-of-the-Art: There is a lack of in-depth comparisons with state-of-the-art methods in compositional text-to-image generation, leaving uncertainties about the proposed method's superiority.

- (Minor) Incomplete Explanations: Some figures and examples, such as those in Figure 5, could benefit from further clarification. The cause of artifacts in certain images and the rationale behind certain choices are not fully explained.

**Justification For Why Not Higher Score:**

**Concerns are not addressed**.

This paper receives reject from all reviewers, and the concerns are not addressed. Here are some example: (1) Reviewer 1 criticizes the lack of novelty, similarity to existing works, and insufficient experimental study. They suggest that the contribution is not clear, and the paper lacks in-depth analysis of the high-level pipeline, questioning the accuracy of the LVLM model.
(2) Reviewer 2 commends the concept of integrating complex models but raises concerns about the unclear contributions of each stage in the pipeline. They emphasize the need for a comprehensive evaluation, including an in-depth ablation study to discern the impact of each component. (3) Reviewer 3 acknowledges the attempt to address an important problem but criticizes the lack of novelty, clarity, and depth in the paper's contributions. They highlight the extensive computational demands of the proposed pipeline and recommend more discussion on algorithm complexity and performance comparisons. (4) Reviewer 4 recognizes the importance of the addressed problem but questions the novelty of the method. They criticize the lack of in-depth comparisons with state-of-the-art methods and suggest more space devoted to algorithm complexity and performance comparisons.

**How to Improve?**

- Clear Roadmap for Improvement: The paper lacks guidance on addressing issues and improving the system when errors occur. A strategy for remediation is essential for users facing challenges with the proposed method.

- Quantitative Analysis of Component Contributions: The individual contributions of each component to the overall system are unclear. A quantitative analysis or test set for evaluation could provide insights into the accuracy and impact of each module.

- Exploration of Alternative Configurations: Exploring alternative configurations, such as substituting LVLM-guided editing with other models, could reveal potential trade-offs in accuracy and response time.

- Comparison with Relevant Methodologies: The paper could benefit from a comparison with relevant methodologies, such as "InstructPix2Pix," "Prompt-to-Prompt," and others mentioned by the reviewers, to demonstrate its advantages and potential alternatives.

- Discussion of Failure Cases: While promising results are highlighted, a discussion of failure cases and system limitations would provide a more comprehensive view of the proposed pipeline's capabilities.

- Scalability to Challenging Scenarios: The paper primarily focuses on simple editing cases. A discussion or experimentation on the pipeline's scalability to more challenging scenarios could strengthen its applicability.

**Summary**

While the paper has strengths in clarity, a conceptual integration of models, and some experimental demonstration, there are significant concerns about novelty, evaluation rigor, and clarity regarding contributions. The reviewers suggest addressing these issues for a more impactful paper.

**Justification For Why Not Lower Score:**

N/A

---

### Decision · Program_Chairs · 2024-01-16

Reject